# Full-field thermal imaging of quasiballistic crosstalk reduction in nanoscale devices

Amirkoushyar Ziabari [1,2], Pol Torres[3], Bjorn Vermeersch[4], Yi Xuan[1], Xavier Cartoixà [3], Alvar Torelló[3], Je-Hyeong Bahk[1,5], Yee Rui Koh[1,2], Maryam Parsa[2], Peide D. Ye[1,2], F. Xavier Alvarez[3] & Ali Shakouri[1,2]

Understanding nanoscale thermal transport is of substantial importance for designing contemporary semiconductor technologies. Heat removal from small sources is well established to be severely impeded compared to diffusive predictions due to the ballistic nature of the dominant heat carriers. Experimental observations are commonly interpreted through a reduction of effective thermal conductivity, even though most measurements only probe a single aggregate thermal metric. Here, we employ thermoreflectance thermal imaging to directly visualise the 2D temperature field produced by localised heat sources on InGaAs with characteristic widths down to 100 nm. Besides displaying effective thermal performance reductions up to 50% at the active junctions in agreement with prior studies, our steady-state thermal images reveal that, remarkably, 1–3 μm adjacent to submicron devices the crosstalk is actually reduced by up to fourfold. Submicrosecond transient imaging additionally shows responses to be faster than conventionally predicted. A possible explanation based on hydrodynamic heat transport, and some open questions, are discussed.

[1] Birck Nanotechnology Center, Purdue University, West Lafayette, IN 47907, USA. [2] Electrical and Computer Engineering, Purdue University, West Lafayette, IN 47907, USA. [3] Departament de Física, Universitat Autònoma de Barcelona, Bellaterra, 08193, Spain. [4] Commissariat à l'énergie atomique (CEA), Le Laboratoire d'Innovation pour les Technologies des Energies Nouvelles et les nanomatériaux (LITEN), 17 Rue des Martyrs, Grenoble, 38054, France. [5] Department of Mechanical and Materials Engineering, University of Cincinnati, Cincinnati, OH 45221, USA. Correspondence and requests for materials should be addressed to A.S. (email: Shakouri@purdue.edu)

S tudy of thermal transport is a crucial part of electronic device design and optimisation[1]. The governing law of heat conduction, Fourier equation, has been the subject of scrutiny as the size of electronic and optoelectronic devices has reached submicron scales. It has been predicted and shown experimentally that as the thermal transport length scales are reduced to the order of the phonon mean free paths, Fourier diffusion equation fails to explain the thermal behaviour[2–5] and thermal transport become non-diffusive.

Non-diffusive thermal transport in semiconductor materials has been extensively studied over the past decade[6–12]. The quasiballistic effects that inherently emerge when the characteristic length scale of the thermal field becomes comparable to that of the dominant heat carriers can be well understood from the Boltzmann transport equation (BTE) for phonons[11,13]. Some recent studies[14,15] have incorporated the non-diffusive characteristic of the heat flow explicitly into their analysis of experimental observations, enabling them to characterise Lévy superdiffusion in semiconductor alloys[14] and phonon transmission spectra of metal/semiconductor interfaces[15]. Most measurements, however, are fitted to a modified Fourier theory that uses a reduced effective semiconductor thermal conductivity[16–20]. These analyses imply that small heat sources will get hotter than nominally expected for a given power density, underlining the added challenges posed by thermal management at the nanoscale, but do little justice to the underlying non-diffusive heat flow physics[7,21].

Moreover, translating the currently available experimental results to a quantitative assessment of thermal performance in actual devices is not straightforward due to several particularities of the utilised thermal metrology techniques. First, they rarely measure the actual temperature distribution, but rather infer quasiballistic deviations indirectly from an aggregate thermal metric. For example, transient thermal grating (TTG) experiments monitor the peak-to-valley contrast of the spatially periodic temperature field[22], whereas pulsed/modulated laser thermoreflectance (TR) signals constitute a spatially averaged surface thermal response due to the Gaussian-shaped probe beam[16,23]. Second, the techniques impose heat sources with extensive temporal bandwidths. This complicates the signal analysis due to non-equilibrium effects, such as electron-hole pair relaxations in TTG[22] and photon–electron–lattice coupling in TR metal transducers[24,25]. Third, the techniques typically operate in quasi-1D heat flow regimes[9,13,16]. Extrapolating the resulting insights to multidimensional configurations is far from trivial, as highlighted by recent TR beam offset experiments that suggested the breakdown of diffusive theory is highly anisotropic[24]. Overall, the question still remains how the observed behaviour affects the thermal field in realistic nanoscale device geometries.

In this work, using full-field thermoreflectance thermal imaging (TRI)[26,27], we try to provide some answers by directly visualising the steady-state and transient quasiballistic thermal fields of gold nanoheater lines fabricated on InGaAs substrate. Figure 1a provide a schematic of TRI setup (more details in methods, and Supplementary Note 1). As we reduce the width of heat sources, the measured temperature of the heater lines exceed those predicted by the Fourier diffusive heat equation. In addition, the non-diffusive behaviour is very significant in the thermal images within 1–3 μm outside the nanoheater lines. Detailed analyses demonstrate that using existing models including Fourier theory with a modified thermal conductivity of the thin film, Fourier theory assuming an anisotropic thermal conductivity[24], as well as incorporating thermal boundary resistance (TBR) between heat source and the substrate along with modified thermal conductivity[19], are not sufficient to consistently explain the full temperature distribution of all device sizes. We show that

a hydrodynamic model is a possible alternative to describe the full-field temperature distribution of all the heater lines under study.

## Results

**Steady-state TR imaging.** Owing to illumination in the visible spectrum and in-situ calibration, TRI provides 2D maps of the absolute temperature rise with spatial resolutions far superior to IR metrology (see Methods). We used electron beam lithography to fabricate a series of samples on a wafer consisting of a 5 μm $In_{0.53}Ga_{0.47}As$ film grown atop InP substrate (see Methods). This material system was chosen based on its relevance to (opto)electronic device technologies aimed at carrying Moore's law deep into the nanoscale[28] and fibre optic telecommunication systems[29,30]. The test devices themselves consist of pairs of gold striplines, one of which is operated as heater, whereas the other can be used as optothermal sensor to monitor the thermal field adjacent to the active junction (Fig. 1b, c). An ultrathin $Al_2O_3$ layer, grown by atomic layer deposition, ensures excellent electrical insulation between the metallisation and semiconductor wafer while adding minimal thermal resistance. We fabricated a variety of heater electrode ranging in widths from 100 nm to 10 μm to allow systematic study of the thermal performance with respect to characteristic device dimension. TR Temperature profiles of a 10 μm and a 400 nm heater line are plotted in Fig. 2a, b.

Besides being far more representative of actual integrated devices than the optically biased structures used in conventional metrology, our electrically operated heater lines offer two additional advantages. First and foremost, the active junction itself can serve as electrical temperature sensor by performing four-probe resistivity measurements (see Supplementary Note 3). Electrically measured junction temperatures closely agree with those inferred from the thermoreflectance images over a wide bias range in both large and small devices. This is shown for a 10 μm and a 400 nm heater line in Fig. 2c, d, respectively. A second benefit is that the heat source is defined much more precisely, since dissipation is completely confined by the current flow in metallisation traces and easily quantified from the supplied bias power. We carried out 3D finite element modelling (FEM) employing material thermal properties measured by TDTR and 3ω techniques (see Supplementary Note 2). The simulated thermal fields are then compared to our experimental images to detect and quantify quasiballistic deviations.

We first turn our attention to the junction temperatures (Fig. 2). For wide heater lines, the measured profiles show a very close match with nominal FEM predictions, both under steady-state (Fig. 2e) and transient-pulsed (Fig. 2g) operations. This observation confirms the accuracy of the FEM diffusive simulations at sufficiently large characteristic length scales. This is clearly no longer the case for smaller devices. Measured junction temperatures in the shown 400 nm wide line, for example, exceed nominal predictions by about 18% (Fig. 2f, h). In analogy with prior literature, one may attempt to interpret the excessive device heating within the modified Fourier framework. We find that an effective InGaAs thermal conductivity reduced by 22% (from 5.4 to 4.2 W m$^{-1}$ K$^{-1}$) indeed fits the measured junction temperatures quite accurately (Fig. 2f, h).

Close investigation of the full-field temperature distribution (Fig. 3a, b) reveals that the modified Fourier profiles with suitably adjusted effective conductivity so as to match the measured temperatures at the active junction (Fig. 3c, d) substantially overestimate the experimentally observed temperature a short distance away from submicron heater lines (Fig. 3e). Remarkably, the recorded temperature tails for the narrowest heater lines ($W$ lower than 300 nm) remain lower than expected even when

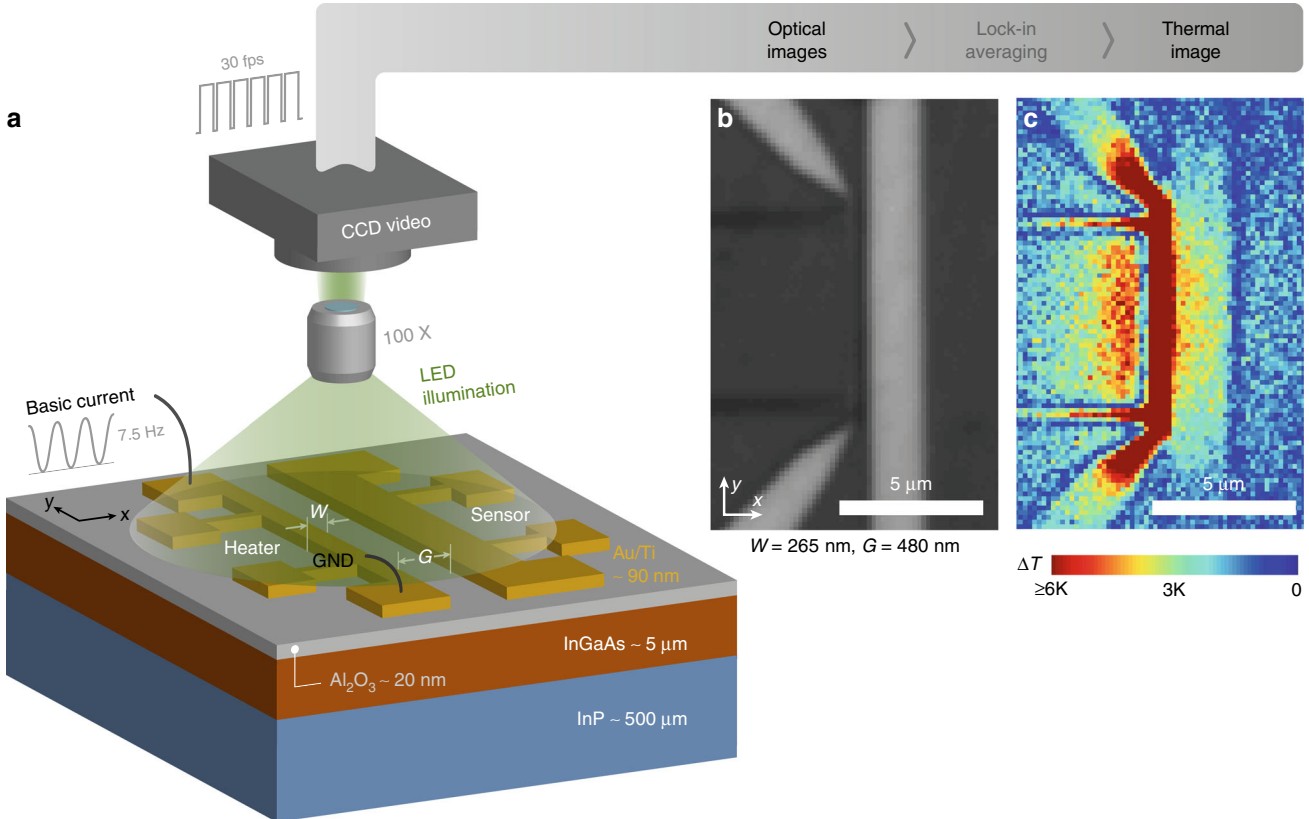

**Fig. 1** Full-field thermoreflectance thermal imaging of quasiballistic transport in InGaAs devices. The test structures **a** consist of electrically biased nanofabricated heater lines of varied widths $W$. A second device, offset by gap distance $G$, serves as thermoreflective and resistive sensor of the thermal field around the heater. A greyscale video camera, phase-locked to the bias signal, gathers optical microscope images **b** under green LED illumination. Averaging the thermally induced brightness variations reveals the 2D surface temperature distribution **c** with ~0.1 K resolution (the plotting range was purposely capped to clearly visualise the lateral decay adjacent to the active junction)

compared to nominal FEM predictions. These findings indicate that the thermal crosstalk adjacent to the submicron devices is reduced.

Analysing Figs. 2 and 3 (and the Supplementary Note 4 on the full set of fabricated samples) shows that as device size reduces, self-heating at the heater line becomes stronger while thermal crosstalk with the neighbouring thermometer is reduced significantly.

**Experimental validation**. We designed a set of measurements to further validate the experimental results. First, a set of identical nanoheater lines were fabricated on top of 3.5 μm thick SiO₂ on a silicon substrate to serve as control samples. The results for three device sizes of 1 μm, 500 nm, and 300 nm are shown in Supplementary Note 6 (see Supplementary Fig. 10). Fourier finite element modelling with a single pair of thermal conductivity of oxide and boundary resistance can explain the full temperature distribution of three device sizes suggesting that the observations made in Figs. 2 and 3 do not appear in the control sample, and thus, they are not artefacts of the TR measurement technique.

Second, to verify that optical effects, such as diffraction, do not impact the comparison between theory and experiment on the thermometer line, temperature measurements were performed at three different wavelengths in the visible range. The results are summarised in Supplementary Note 7. TR measurements performed at 470, 530 and 660 nm wavelengths (see Supplementary Fig. 11) show consistent behaviour on the junction as well as on the neighbouring thermometer, suggesting that the observations made in Figs. 2 and 3 are not due to optical effects.

Third, we investigated the impact of temperature-dependent thermal conductivity of InGaAs. The results are shown in Supplementary Note 8. Temperature profile of a pair of 10 μm heater lines at multiple power density inputs were obtained. It is shown (see Supplementary Fig. 12) that the temperature profile scales linearly with the power values both on top of the heater line as well as on the neighbouring line. This in turn demonstrate that the impact of the temperature dependence of the InGaAs thermal conductivity is negligible in the range of localised heating studied in this work. Additionally, as it is evident from Fig. 3b, d, whereas the temperature change at the top is about 35 K, its value on the tail is only about 4 K, suggesting that the temperature change across the InGaAs thin film is small and will not impact the measured temperature. This is because the temperature-dependent thermal conductivity of InGaAs in 300–340 K range is negligible (<5% change based on the TDTR data)[31].

**Transient TR imaging**. The transient TR imaging responses of the heater lines also show discrepancies between the experiment and the modified Fourier model. This is shown in Fig. 4. The temperature evolution of the heater line over time in response to a 1 μs electrical pulse with 10% duty cycle is plotted in Fig. 4a–c for 500, 400 and 200 nm heater lines, respectively. In each panel the maximum TR temperature change on top of the heater line is plotted against time. The evolution in time of TR temperature change is compared with the corresponding modified Fourier model results (cyan line in the figure). The same thermal conductivity as their steady-state values are used for each heater line (Fig. 3e). It is evident that the experimental transient responses of

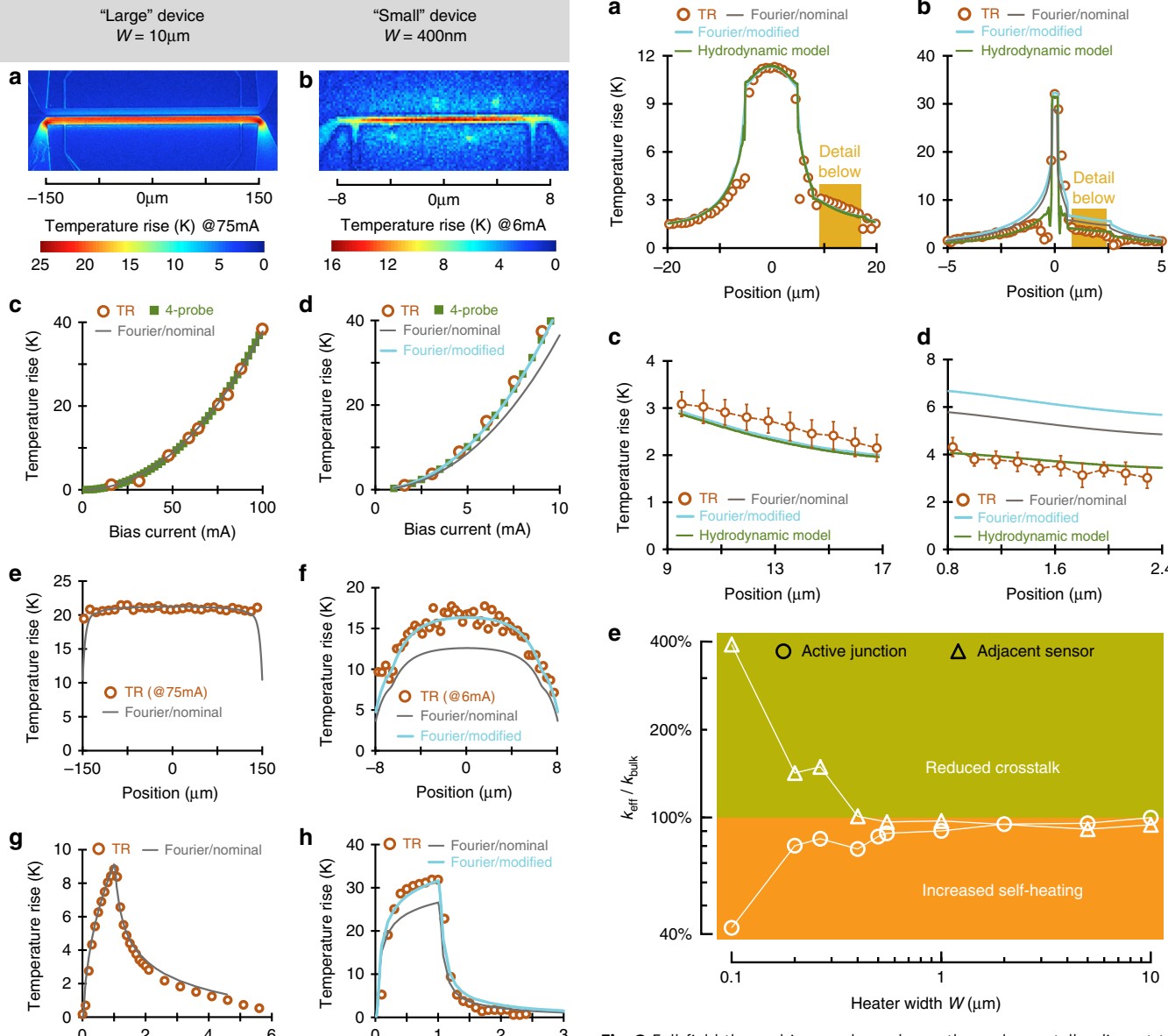

**Fig. 2** Characterisation of thermal performance at the active junction. Thermoreflectance thermal images of **a** a 10 μm; and **b** a 400 nm heater lines. The average device temperature on top of the devices for **c** 10 μm; and **d** 400 nm heater lines. The thermoreflectance, electrical (IVT) and FEM results are plotted as a function electrical current. FEM results with nominal value of thermal conductivity for InGaAs (5.4 W m⁻¹K⁻¹) agrees with the experimental results at 10 μm, whereas a smaller number (4.2 W m⁻¹K⁻¹) is needed to match the data for 400 nm device. Comparison of TR and FEMs' longitudinal temperature cross sections along **e** 10 μm; and **f** 400 nm heater lines. Comparison of maximum transient temperature on top of the heater line using TRI and FEM: **g** 10 μm; and **h** 400 nm heater lines. A 1 μs current pulse with 5% duty cycle was applied to the heater and the temperature profile was recorded every 100 ns. The average temperature near the centre of the line is compared to that of obtained from FEM. Marked agreement between the FEM model using nominal value of thermal conductivity (5.4 W m⁻¹K⁻¹) and the experiment in all of the results for 10 μm heater line is evident. The measured self-heating in small devices clearly exceeds nominal predictions but can still be fitted to diffusive simulations with reduced InGaAs thermal conductivity (4.2 W m⁻¹K⁻¹ instead of 5.4 W m⁻¹K⁻¹)

**Fig. 3** Full-field thermal image shows lower thermal crosstalk adjacent to nanoscale junctions. Thermal images (as those shown in Fig. 2a, b) also yield the temperature field perpendicular to the heater lines (perpendicular cross section) for **a** a 10 μm; and **b** a 265 nm heater line. The expanded view of temperature cross section (highlighted in the yellow boxes) is shown in part **c** for a 10 μm; and **d** for a 265 nm heater line. Each data point is obtained by averaging few neighbouring pixels along horizontal axis, and the errorbars are the standard deviation of those pixels. It is evident that while the junction temperature can be accurately fitted by conventional interpretations with reduced InGaAs thermal conductivity, these notably overpredict the thermal field observed adjacent to small devices. The effective InGaAs thermal conductivity (normalised here to its nominal value $\kappa_{bulk}$) required in diffusive simulations to fit the measured temperature profile at the indicated locations is strongly dependent on characteristic dimension. This is shown in **e**. $\kappa_{eff}$ is defined as the isotropic value needed in the Fourier model to match the measured temperatures at the heater or the thermometer. Although thermal transport around larger devices ($W$ greater than 1 μm) obeys nominal Fourier theory almost perfectly, increasingly non-Fourier thermal behaviour is observed at/nearby the active junctions of submicron devices

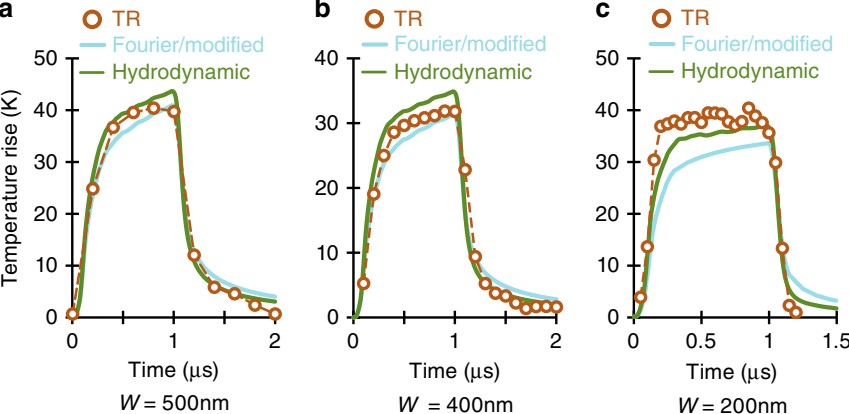

**Fig. 4** Temporal failure of modified Fourier theory. Transient responses for small devices with line widths of **a** 500 nm, **b** 400 nm and **c** 200 nm. Finite element Fourier model (grey lines) with adjusted InGaAs thermal conductivities that accurately capture the quasi-static peak temperatures of each device do not match the faster heating/cooling time constants of the experimental data. Hydrodynamic (KCM) capture better the faster transient response, but minor discrepancies still exit

the 400 and 200 nm heater lines do not follow the modified Fourier model. Interestingly, both the heating and cooling rates of the experimental data are faster than the Fourier simulations. The effect is more prominent as the device width decreases.

## Discussion

The strongly contrasting InGaAs thermal conductivity values required in the FEM simulations to reproduce distinct portions of the measured temperature field show that modified Fourier theory fails to properly capture the quasiballistic transport physics. Beside modified Fourier, we investigated alternative approaches in literature, namely using an anisotropic thermal conductivity[24] and incorporating thermal boundary resistance (TBR) between heat source and the substrate along with modified thermal conductivity[19] to address the departure from diffusive heat transport. The details are shown in Supplementary Notes 5.1–5.3.

Anisotropic Fourier theory has several limitations to explain all of the experimental results (Supplementary Note 5.2 and Supplementary Fig. 5). First of all, it is hard to justify an anisotropic thermal conductivity of 1.5 W m$^{-1}$ k$^{-1}$ in-plane and 10 W m$^{-1}$ k$^{-1}$ cross-plane for a five micron thick InGaAs. Second, for each device size a new pair of anisotropic thermal conductivity values is required to fit the temperature profiles. Also, for very narrow heaters, some disagreements still remain on the tails of the temperature profile (Supplementary Fig. 5c).

Separating TBR from the oxide and substrate thermal conductivities was also investigated using finite element model (see Supplementary Note 5.3). It is noticed that although one could fit the entire temperature profile (see Supplementary Fig. 6), a distinct pair of thermal conductivity and TBR was needed for each device size (the fitted TBR value was not constant). Supplementary Table 1 summarises pair of conductivity and TBR values needed to fit experimental results of different device widths from 10 μm to 200 nm. Not only the pairs are distinct for each device size, the values obtained for TBR are very large compared with the known values of TBR based on the TDTR characterisation or diffuse or acoustic mismatch models (DMM and AMM), which are on the order of 1–5 nK m$^2$ W$^{-1}$[32,33].

It was recently shown that superdiffusive Lévy transport in alloys can explain the reduced apparent thermal conductivity at the source[7]. The tempered Lévy model is validated in Supplementary Fig. 7. As described in Supplementary Note 5.4, this model predicts the correct trend at the source (increased self-heating and corresponding higher temperature at the heater line),

but a near nominal or even increased thermal crosstalk (and corresponding higher temperature) at the nearby thermometer (see Supplementary Fig. 8), which contradicts the lower thermal crosstalk observed in the experiments. This failure is a direct and inevitable consequence of the fact that the associated Green's function nowhere crosses over with the diffusive counterpart. One should note that this failure would equally occur in BTE analysis within the relaxation time approximation (RTA). Indeed, the rigorous single pulse response of the 1D RTA-BTE invariably displays impeded thermal transport that gradually recovers (but never trumps) diffusive behaviour[13]. Explaining the present measurement therefore clearly requires additional physics going beyond the current state-of-the-art RTA/superdiffusion formalisms.

Here, we provide a possible explanation based on a hydrodynamic model (see Methods) that provides good agreement with our experimental results with minimum number of fitted parameters. The approach rests upon the following relation between heat flux **q** and temperature gradient $\nabla T$ that can be derived directly from the BTE (Torres et al., in preparation):

$$\mathbf{q} - \left[\ell^2\left(\nabla^2\mathbf{q} + 2\nabla\nabla \cdot \mathbf{q}\right)\right] = -\kappa\nabla T. \tag{1}$$

Here $T$ is temperature, **q** is the heat flux, $\kappa$ is the thermal conductivity and $\ell$ a characteristic length scale. The details of this model and an analysis of some of its consequences are described elsewhere (Torres et al., in preparation). The term in square brackets accounts for nonlocality induced by non-resistive normal phonon scattering processes and gives rise to hydrodynamic (fluid-like) effects over characteristic length scale $\ell$. This beyond Fourier correction has a marked impact on the thermal fields in the vicinity of small sources. In particular, our measurements can be well understood qualitatively in terms of the notable vorticity that emerges near the edge of the heater line (Fig. 5). Here the flux field bends over a region with characteristic size $\ell$ and is no longer parallel to the thermal gradient, thereby achieving additional cooling of the semiconductor surface in the area adjacent to the heater line. Notice that the effect of the Laplacian cannot be absorbed in an effective thermal conductivity due to its intrinsic non-isotropical nature. In Fig. 5, the vorticity is only shown for few hundreds of nanometre in the vicinity of nanoheater lines. Vorticity, however, extends few micron away from nanoheater line which reaches the area below the thermometer line (see Supplementary Note 9 and Supplementary Fig. 13).

Quantitatively, FEM simulations of semi-infinite InGaAs substrates governed by constitutive Eq. (1) with a single hydrodynamic length $\ell = 150$ nm closely match the entire measured temperature field (both at and around the active junction) for all investigated device sizes (Fig. 3a, b, and Supplementary Fig. 4). The nonlocal term also provides a better fit of the transient experiments (Fig. 4a, b). Some differences in the amplitude of transient response at early times still exist (Fig. 4c).

It may seem somewhat surprising to invoke hydrodynamic transport in InGaAs. After all, the bulk conductivity of this alloy material is dominated by mass impurity scattering, and routinely predicted within the relaxation time approximation which treats normal processes simply as fully resistive. However, bulk conductivity is inherently evaluated under the assumption of a 1D temperature gradient, whereas the vectorial nature of hydrodynamic vorticity only reveals itself in multidimensional settings.

Moreover, vorticity only becomes important when the characteristic source dimension becomes comparable with the hydrodynamic length $\ell \sim 150$ nm, being about an order of magnitude smaller than the dominant phonon mean free paths that govern bulk transport.

In conclusion, full-field thermoreflectance thermal imaging of InGaAs devices yielded a detailed characterisation of nanoscale thermal transport beyond those achievable in conventional metrology. The observation of lower thermal crosstalk adjacent to submicron active junctions and faster transient response at the heat source are particularly remarkable, as such unexpected behaviour would normally not be associated with the necessary evil of quasiballistic heat flow effects. Our findings imply promising prospects for reduced thermal crosstalk between neighbouring transistor channels or photonic arrays in nanoscale systems.

## Methods

**Thermoreflectance thermal imaging**. Thermoreflectance (TR) thermal imaging is based on the change in materials' reflection coefficient as a function of temperature. For quasi steady-state measurement, we use a low-frequency (7.5 Hz) sinusoidal electrical current. A constant LED light is illuminated on the device under test (DUT), while the device is being biased. Typically, 530 nm illumination wavelength is used for gold samples but the wavelength can be tuned to maximise the signal. The reflected light from the DUT is captured by a CCD camera with synchronous lock-in detection. Using the calibrated coefficient of thermoreflectance ($C_{TR}$), temperature profile is extracted. Typical sensitivity is on the order of 0.1 °C for 5 min averaging. By pulsing the LED light and with the use of boxcar averaging, transient thermal images with 50 ns resolution can obtained. More details are described in the Supplementary Note 1 and Supplementary Fig. 1.

**Device fabrication**. The native oxide on the $In_{0.53}Ga_{0.47}As$ (5 µm)/$In_{0.52}Al_{0.48}As$ (100 nm)/InP (500 µm) sample (grown by molecular beam epitaxy) was removed with 1 min dilute HF solution dip. Subsequently, a 20 nm $Al_2O_3$ insulation layer was deposited using the atomic layer deposition (ALD) technique at 200 °C followed by rapid thermal annealing at 450 °C for 30 s. Au heater lines were then fabricated using a Ti (~5 nm) adhesion layer through electron beam lithography (EBL), metallisation and lift-off. The Au thickness was measured to be 90 nm. The aspect (length-to-width) ratio of each device was fixed to 40 (except for 10 µm, for which the aspect ratio was 30). Four large contact pads, each $80 \times 80$ µm², were fabricated for each heater line, so that the samples can be probed easily and also the thermal measurement can be further confirmed using electrical measurement of the heater resistance. Two similar heater lines were placed in parallel next to each other with distances (gap sizes) of 300 nm, 500 nm and 20 µm. In this case one of the heater line works as a heater and the other heater line serve as thermometer.

**Finite element modelling**. ANSYS finite element modelling (FEM) was used to calculate full 3D steady-state and transient temperature profile in the devices. Material parameters were obtained from TDTR, 3ω, and temperature-dependent current-voltage (IVT) measurements. These parameters are provided in steady-state results section as well as in Supplementary Note 2 and Supplementary Fig. 2. Electrical and thermal conductivities, as well as temperature coefficient of resistance (TCR) of gold lines are individually calibrated using the IVT measurements (see Supplementary Note 3). There is a minor width dependence compared to the bulk gold properties as described in Supplementary Fig. 3. This can be due to the metal line grain boundaries and edge roughness that have a more important role in narrower heaters. Heat capacity and mass density for transient modelling were set according to literature values for Au, $Al_2O_3$, InGaAs and InP. Over a million mesh elements were used in the full 3D FEM model to ensure accuracy of the 3D FEM Model.

**Hydrodynamic model**. This is based on the kinetic-collective model (KCM) that takes into account the non-resistive normal scattering as well as the nonlocal effects due to the appearance of phonons with large mean free paths. In this model, two different transport regimes are considered, a kinetic regime where all phonons are independent and a collective regime where a group of phonons share the same velocity and mean free paths. Collective phonons give rise to fluid-like hydrodynamic transport. KCM has been used to predict the thermal conductivity of bulk media, thin films and nanowires using scattering terms from ab-initio calculations[34]. From the Boltzmann transport equation (BTE), one can derive the hydrodynamic Eq. (1) that includes memory and nonlocal effects. The details of this model and an analysis of some of its consequences are described elsewhere (Torres et al., in preparation). Solutions to BTE taking into account the full phonon spectrum for complex multilayer and 3D geometries, such as that in our experiment, are computationally prohibitive. However, Eq. (1) can be easily integrated

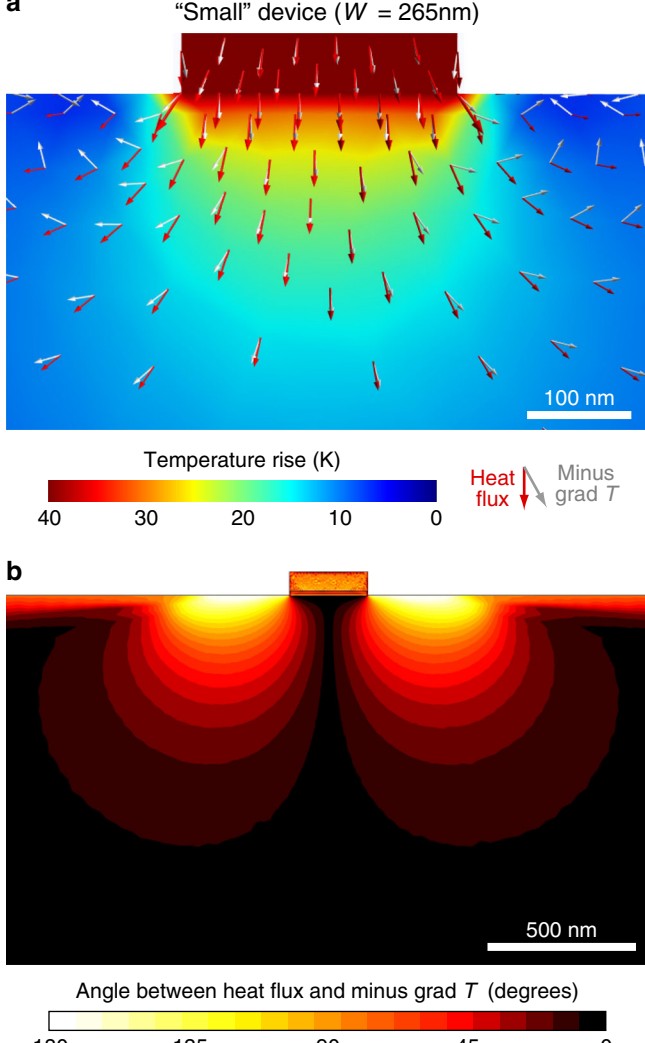

**a** "Small" device ($W = 265$nm)

Temperature rise (K)

| 40 | 30 | 20 | 10 | 0 |

Heat flux ↓   Minus grad $T$

**b**

Angle between heat flux and minus grad $T$ (degrees)

| 180 | 135 | 90 | 45 | 0 |

Antiparallel   Orthogonal   Aligned

**Fig. 5** Fluid-like thermal transport in room-temperature solid media. Hydrodynamic model simulations of the steady-state temperature field (**a**) are in good agreement with the measured surface temperature profiles both at and nearby the active device (see Fig. 3a–d). The hydrodynamic heat flux and temperature gradient are found to be severely misaligned adjacent to small sources. **b** It is this vorticity that may be physically responsible for the experimentally observed disagreement in effective thermal performance over those regions (see Fig. 3e)

with the help of finite element method. Despite its apparent simplicity, the vectorial characteristic of Eq. (1) includes 3D hydrodynamic effects in addition to the intrinsic thermal diffusion (Torres et al., in preparation). This is also consistent with the theoretical modelling of Ramu and Bowers[10].

**Data availability**. The authors declare that the data supporting the findings of this study are available within the article and its Supplementary Information files or from the corresponding authors on reasonable request.

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

## Acknowledgements

A.Z. would like to thank Prof. Mark Lundstrom and Dr. Xufeng Wang for their insightful comments and discussions. B.V. acknowledges funding from the ALMA project (European Commission Grant No. 645776). P.T., A.T., F. X.A. and X.C. would like to acknowledge Profs. Juan Camacho, Javier Bafaluy and David Jou for useful discussions regarding hydrodynamic phonon transport. P.T., F.X.A. and X.C. also acknowledge the financial support of the Spanish Ministry of Economy and Competitiveness under grants TEC2015-67462-C2-2-R (MINECO/FEDER) and TEC2015-67462-C2-1-R (MINECO/FEDER), and the support of the Department d'Universitats, Recerca i Societat de la Informació (DURSI) of the Generalitat de Catalunya under grant 2014-SGR-384.

## Author contributions

A.Z., B.V., J.H.-B. and A.S. designed the experiments. Y.X., J.H.-B. and P.Y. fabricated the devices. A.Z. performed TR, IVT, and 3ω experiments, analysed the data and performed numerical FEM simulations and calculations. Y.K. did the TDTR measurements and analyses. M.P. performed TR experiments and helped with manuscript preparation. P.T., X.C., A.T. and F.X.A. carried out the hydrodynamic numerical simulations and analyses. A.Z., B.V., P.T., F.X.A. and A.S. discussed the results. A.Z., B.V. and A.S. wrote the manuscript text.

## Additional information

**Competing interests:** The authors declare no competing financial interests.

