## [Peer Review File · Nature Communications]

Reviewers' Comments:

Reviewer #1:

Remarks to the Author:

The authors describe their experiments on the temperature response of InGaAs to heating from a metal line whose width is varied from 10 microns to 265 nm. By using the thermorefectance of a Au film adjacent to the heater, the authors examine the temperature profile variation at the surface of the InGaAs near the heater line. As expected, when the dimensions of the heating line become small, predictions based on the Fourier equation break down. They fit their data with a hydrodynamic model for heat-conduction, and conclude the thermal transport is "enhanced" due to momentum conserving normal phonon scattering. Hydrodynamic flow is not usually important in transport problems because defect or Umklapp scattering prevent flow. The authors assert that this is no longer true on length-scales of 100 nm. I think the idea of hydrodynamic transport being important on sub-micron length scales is interesting, and may influence thinking in the field. However, I do not recommend for publication in Nature Communications because the experiments are not thoroughly validated, and the conclusions are not justified by the data. The authors do not make a compelling case that the heat-current is enhanced in their devices, and do not validate their claims of hydrodynamic thermal transport through a systematic test of their hypothesis.

(1) As the author's note, their work is one of many observations of a breakdown in the heat-diffusion equation for small heating lines. Understanding and being able to predict the thermal response of a nanoscale system is technologically important and is therefore of interest to a broad community. But a troubling pattern exists in papers describing the breakdown in Fourier theory, and this work appears to be following that pattern. Every new observation of a breakdown in Fourier theory comes attached with a new model, or a new explanation. In this manuscript, another new model is proposed, a hydrodynamic model for thermal transport. No explanation is given for why old models are inadequate or failed.

(2) Many of the present authors have published a lot of interesting work on how thermal transport models based on superdiffusion can explain a breakdown of Fourier theory in 1-dimensional semiconductor alloys. Here, they don't analyze their data in terms of superdiffusion at all, instead choosing to introduce a new "hydrodynamic" model. The test of a good model is not whether it can explain data. It's whether it can make predictions. Moving from a 1-dimensional to 3-dimensional heat-transfer problem would seem to be a nice test of the superdiffusive transport regimes the authors have previously championed. I assume since a new model is presented that is unrelated to superdiffusion, the superdiffusive model can't explain this experiment. Is that right?

(3) Experimental evidence of hydrodynamic transport would be an important advance. But the current results are only mildly suggestive because no systematic study is conducted that tests the role of normal phonon-phonon scattering in nanoscale thermal transport.

(4) The hydrodynamic model appears to only predict vorticity within a few hundred nanometers of the heater line (Fig. 5). Because the Au temperature sensor cannot be perfectly adjacent to the heater line, this is exactly the spatial region the authors are not probing.

(5) The authors don't validate their technique with a control experiment. They should show Fourier works for 255 nm heater lines on a material with a similar thermal conductivity in which they don't expect "quasiballistic transport"

(6) The authors observe that the in-plane temperature profile is smaller than what the heat-equation predicts. This is consistent with the observations made in TDTR experiments from Cahill's group, see Fig. 1 of Ref. 24. The authors claim this is evidence of an "enhancement" of the heat-flow. I'm not entirely clear on what they mean by "enhanced". Experimentally, they only know the total heat-current out of the heater line, and the temperature profile perpendicular to metal line at the surface of the InGaAs. I don't see a way to know whether heat is travelling parallel or

perpendicular to the surface. In other words, they aren't measuring an in-plane heat-current, so it's unclear how they can conclude the in-plane heat current is greater or less than what Fourier theory would predict for the in-plane temperature profile they observe.

(7) Looking at Fig. 5, it's difficult to see evidence of "enhancement" in the model calculation. In some positions, the heat-current is through-plane while the temperature gradient is in-plane. That seems like a suppression of the in-plane heat current. In other places, the heat-current is in-plane while the temperature-gradient is through-plane. I guess in those locations the in-plane heat-current is "enhanced". But on net, the heat-current must be suppressed, otherwise the temperature of the heater line would be lower than Fourier equation predicts, not higher.

(8) As the authors note in their supplemental information, recent work from both Cahill's group at Illinois, and Minnich's group at CalTech suggests that interfacial scattering impacts thermal transport in these types of problems. It doesn't appear the authors consider the role of the Au film or the interface in their experiments. How does the Hydrodynamic model account for interfacial scattering and the Au film? Since the transport is nonlocal, I don't think transport in the InGaAs can be treated independently from the transport across the Au/InGaAs interface.

(9) The thermoreflectance value at 530 nm is determined by the electronic band structure of Au. The thermoreflectance near 530 nm in Au is high due to a d-band/s-band transition, see for example Eesley PRB 2144 (1986). The authors report a significant decrease in the thermoreflectance of the Au when they make the wire thin. It's difficult to see how 100 nm scale dimensions in the wire would affect the electronic structure of Au. It's more likely to be some sort of artifact related to the diffraction limit.

In summary, while the main claim of the paper of "enhanced" heat transfer in a nanoscale device due to hydrodynamic heat transfer is interesting, I do not find the evidence provided by the authors convincing of their claim. To show heat transfer is enhanced in the in-plane direction would require a measure of the in-plane heat-current and the in-plane temperature profile. To show hydrodynamic transport is important, the authors would need to do a systematic study where something like the ambient temperature or material is changed to alter the importance of normal phonon-phonon scattering, and therefore alter the hydrodynamic transport.

Reviewer #2:

Remarks to the Author:

This paper uses a thermoreflectance imaging (TRI) technique to study the nonlocal effect of phonon transport when the heater dimension changes from 10 μm to as small as 265 nm. The main innovation is that unlike recently developed pump-probe techniques which study nonlocal effect by extracting an effective thermal conductivity based on Fourier's law, the TRI method offers a temperature map, which provides more fruitful information beyond a single extracted thermal conductivity value. The major finding is that using an effective thermal conductivity, which accounts for the nonlocal effect, cannot describe the phonon transport and thus temperature distribution correctly. It was claimed that while non-local effect should lead to a lower effective thermal conductivity, non-local region near the heater shows enhanced heat transfer performance, which is counter-intuitive from the conventional non-local transport picture. The authors finally found that using a hydrodynamics model can better describe the temperature profile involving non-local effects. The paper is interesting and presents a new angle to the non-local effect. However, the reviewer has the following comments and hopes the authors to address them. Major modification is needed before the reviewer can recommend this paper for publication.

1. The captions of the figures are not informative. The authors should describe clearly what each panel in a figure means. For example, it is still not completely clear to the reviewer what was plotted in Fig. 4d. There are discussions of information of each panel in the text, but they are still

not clear. The reviewer usually has to read the text many times to understand a figure.

2. In Fig. 3a, for the small device, how to understand the spikes next to the heater from the hydrodynamics model?

3. For the hydrodynamic model, is k isotropic? Will anisotropy influence the results? Why not make it anisotropic as it should be?

4. Fig. 4 shows temperature rise up to ~ 40 K, how will the temperature dependent thermal conductivity influence the temperature profile? Did the authors consider this? Non-local effect should also depend on temperature, since phonon MFP is a function of temperature.

5. The authors claims that Fourier's law with an effective thermal conductivity cannot describe the heat transfer correctly. However, thermal conductivity is a function of temperature as mentioned above and thus location. If such an effect is considered, will Fourier's law be potentially able to describe the heat conduction involving non-local effect? The reviewer feels that the authors should be more cautious in discounting Fourier's law as a way to describe non-local heat conduction, since there are so many studies out there showing that it can do such a job reasonably well.

6. The resolution of the TRI method is 200-300 nm. The smallest device has a feature length of 263 nm, which is similar to the resolution. How would this influence the accuracy of temperature measurement and thus reliability of the data.

In the following we answered the questions asked by the reviewers.

REVIEWER 1:

(1) As the author's note, their work is one of many observations of a breakdown in the heat-diffusion equation for small heating lines. Understanding and being able to predict the thermal response of a nanoscale system is technologically important and is therefore of interest to a broad community. But a troubling pattern exists in papers describing the breakdown in Fourier theory, and this work appears to be following that pattern. Every new observation of a breakdown in Fourier theory comes attached with a new model, or a new explanation. In this manuscript, another new model is proposed, a hydrodynamic model for thermal transport. No explanation is given for why old models are inadequate or failed.

This is a very important comment. We added several paragraphs in the main text (pages 6 and 7), and also added three sections to the supplementary information to address these concerns. Here is a summary:

This work presents a complete 2D temperature map of a heated surface with submicron precision. In most previous studies, thermal responses were obtained using only single point temperature information or averaged temperature measured over area of laser probe. Full field thermal imaging of different heater lines on the same thin film and independent electrical sensor as thermometer allow us to test the validity of previously presented models.

1) Using a bulk effective thermal conductivity of the substrate:

Minnich et al.¹ used a phonon diffusive – ballistic transition to explain their results. The final effect is an effective reduction of the substrate thermal conductivity. This approach was used because their experimental data consisted only in the temperature decay of the heater and consequently only effective information at the source was obtained. Several other authors made the same observation using TDTR technique which are listed in the main text. The temperature profile of a 265nm heater line along with temperature cross sections are shown in Figure R111. The red dots are the experimental results in this figure and the dash lines are the FEM results based on Fourier.

Akin to the prior works in literature, using a reduced thermal conductivity of 4.5W/m-K for InGaAs (black line) we can fit the heater temperature at the top as shown in Figure R111b, and c. However, we observe an overprediction of temperature on the neighbouring thermometer line meaning we cannot explain the entire temperature profile with a reduced thermal conductivity. On the other hand, we can use a larger thermal conductivity of 8W/mK to match the tail of temperature distribution on the thermometer however using that value we cannot match the temperature profile at the top of the device (blue line in Figure R111b and c). In general, changing to an effective thermal

conductivity we can locally match the average temperature as was done in previous studies, but there is not a single effective thermal conductivity that matches the entire temperature profile.

Figure R111. a. temperature profile of a 265nm heater line. b. cross section along the heater line. red dots are the experimental results. black and blue lines are Fourier simulation with isotropic thermal conductivity of 4.5W/m-K and 8W/m-K. Green line is the Fourier results with anisotropic thermal conductivity (10w/m-K cross-plane and 1.5W/m-K in-plane). c. Vertical cross-section compares the experimental and FEM results. Figures b and c shows that a lower conductivity can fit the data at top of the heater line and a larger conductivity could match the data on the tail, however to match the full distribution we needed anisotropic thermal conductivity in the Fourier model which is hard to justify for 5μm thick film. Additionally, the same combination of anisotropic thermal conductivity doesn't fit all the experimental data and only works for 265nm wide heater.

2) Using an anisotropic thermal conductivity.

A similar (but not the same) observation was made by Wilson and Cahill. In their beam-offset TDTR setup an averaged measure of the temperature on the heat source and near the heated regions was obtained. A lower temperature than the one predicted by the effective Fourier was observed. This setup offers the possibility to fit to two points on the profile, using an anisotropic thermal conductivity. The green curve in Figure R111b and c show the results. Although the obtained curve is better than the single parameter approach, there are few issues with this approach:

1. It is hard to justify an anisotropic thermal conductivity of 1.5 W/mK in-plane and 10 W/mK cross-plane for five microns thick InGaAs. Reduction of effective thermal conductivity has been observed for thin film materials, but this is typically for samples that are <0.5-1micron thin. Also, TDTR measurements on the same sample gave InGaAs thermal conductivity of ~6W/mK. It is hard to justify why the cross-plane thermal conductivity could be more than 50% larger and in-plane value 400% smaller.
2. For each nanoheater device size on the same InGaAs substrate a new pair of thermal conductivity values is required to fit the temperature profiles.
3. There are samples for which we could not find a pair of anisotropic thermal conductivity to fit the entire temperature profile
4. Even for the measurement in Figure R111 the thermal conductivity pair does not match the entire temperature profile and disagreement on the tails still exist

3) Using a Thermal boundary resistance.

Several works have interpreted deviations from Fourier diffusion in terms of an effectively increased Thermal Boundary Resistance (TBR) between the heat source and the substrate. For example, Siemens et al² used a constant TBR along with a modified thermal conductivity to explain their experimental results.

Here we took the same approach and used a separate TBR in addition to oxide and substrate layers to explain the experimental results. Our experimental results are summarized in Table R111 and Figure R112. Figure R112 shows the experimental results (blue dots) for 265nm heater line and modelling results with different pairs of TBR and thermal conductivity. It can be seen that including a TBR of 9 nKm²/W (dark red line) we can fit the heater temperature but we obtain poor predictions for the rest of the points. If we also change the thermal conductivity using the two parameter approach to fit to heater and temperature (red line), we can get a curve similar to our hydrodynamic (KCM) approach.

Figure R112. Comparison between experimental and numerical results for horizontal temperature cross section along 265nm heater line. Blue dots are experimental results. Different lines correspond to different pairs of TBR and thermal conductivity. The pairs are changed until the best fit between numerical and the experimental results are obtained.

If we repeat this procedure for the rest of the lines, we get the results summarized in Table R111 as the best fitting results:

Table R111. The best pairs of TBR and thermal conductivity to fit the experimental results at different width

Line	200 nm	400 nm	500 nm	5 μm	10 μm
k (W/m·K)	8.1	5.6	6.1	5.3	5.3
TBR (nKm ² /W)	38.1	40.6	33.7	14.7	9.2

This approach has doubled the number of fitting parameters, the thermal conductivity and the value of the TBR. A distinct pair of values is needed to fit observations at each device dimension. In addition, the main drawback of this approach is that the magnitude of the obtained TBRs are very large compared to the TDTR measurements and the known values of TBR models such as Diffuse or Acoustic Mismatch in the order of some nKm²/W.

In conclusion, a new model to describe experimental results was pursued given the poor predictions given by the existing theories. Using KCM we only need to fit the thermal

conductivity as the other parameter (nonlocal scale parameter) is equal for all the lines ($l=150\text{ nm}$). This also offers a different physical interpretation for the appearance of extra resistivity in small heater lines with the nonlocal vorticity effects.

(2) Many of the present authors have published a lot of interesting work on how thermal transport models based on superdiffusion can explain a breakdown of Fourier theory in 1-dimensional semiconductor alloys. Here, they don't analyze their data in terms of superdiffusion at all, instead choosing to introduce a new "hydrodynamic" model. The test of a good model is not whether it can explain data. It's whether it can make predictions. Moving from a 1-dimensional to 3-dimensional heat-transfer problem would seem to be a nice test of the superdiffusive transport regimes the authors have previously championed. I assume since a new model is presented that is unrelated to superdiffusion, the superdiffusive model can't explain this experiment. Is that right?

We greatly appreciate the Reviewer's awareness of our prior work on superdiffusive Lévy transport in alloys. The question why this was not applied here is both a natural and fair one. In early stages of the work, we did analyse the heater line configuration using a 3D tempered Lévy framework. As summarised below, this approach readily yields the correct trend at the source (increased self-heating and corresponding higher temperature at the heater line), but is unable to reproduce the lower thermal crosstalk (and corresponding lower temperature) measured at the nearby sensor.

The superdiffusive framework has been extended to multidimensional heat flow by one of the authors last Fall³. The evolution of volumetric thermal energy density P in an infinite medium can be described in terms of an isotropic stochastic process. In Fourier-Laplace domain the single pulse response takes the form

$$P(\|\vec{\xi}\|, s) = \frac{1}{s + \psi(\|\vec{\xi}\|)}$$

where $\vec{\xi}$ denotes the spatial frequency vector and s is the Laplace variable.

3D transport in semiconductor alloys is found to obey a tempered Lévy process.

$$\psi(\zeta) = \frac{D\zeta^2}{(1 + r_{LF}^2\zeta^2)^{1-\frac{\alpha}{2}}} \quad \text{where } \zeta \equiv \sqrt{\zeta_x^2 + \zeta_y^2 + \zeta_z^2}$$

At long length scales ($\zeta r_{LF} \ll 1$) this converges to regular diffusion $\psi \simeq D\zeta^2$ with bulk (Fourier) diffusivity D while short length scales ($\zeta r_{LF} \gg 1$) exhibit pure Lévy dynamics $\psi \simeq D_\alpha \zeta^\alpha$ with superdiffusion exponent $1 < \alpha < 2$ and fractional diffusivity $D_{\text{alpha}} \equiv D/r_{LF}^{2-\alpha}$ (unit $\frac{m^\alpha}{s}$). The transition between the two asymptotic regimes takes place over

characteristic length scale r_{LF} . Tempered Lévy analysis of TDTR experiments on an InGaAs sample produced³

$$\alpha = 1.71, r_{LF} = 550\text{nm}$$

We adopted these values alongside bulk diffusivity $D = 3.55\text{mm}^2/\text{s}$ (conductivity $\kappa = 5.5 \frac{\text{W}}{\text{m}\cdot\text{K}}$) in the simulations that follow. We note that the sample used in the present work has a larger film thickness than the one previously measured with TDTR and thus likely possesses slightly different parameter values, but as these deviations do not affect the key outcome in any way they can be safely ignored.

To compute the thermal profile induced by a heater line, we first obtain the steady-state ($s = 0$) Green's function in real space through 3D Fourier inversion:

$$G(r) = \frac{1}{2\pi^2} \int_0^\infty \zeta^2 \frac{1}{\psi(\zeta)} j_0(\zeta r) d\zeta \quad \text{where } j_0(u) = \frac{\sin(u)}{u}$$

The integral must be evaluated numerically. The result, which as expected transitions between pure Lévy and Fourier asymptotes (both of which are available in closed form), can be described compactly as

$$G(r) \simeq \frac{\left[1 + \left(\frac{r_0}{r}\right)^m\right]^{\frac{2-\alpha}{m}}}{4\pi D r}, r_0 = 2 \left[\frac{\Gamma\left(\frac{3-\alpha}{2}\right)}{\sqrt{\pi}\Gamma\left(\frac{\alpha}{2}\right)} \right]^{\frac{1}{2-\alpha}} \cdot r_{LF}$$

For the InGaAs parameters listed earlier, $m=1.62$ provides the best fit (deviations $\leq 1.5\%$)

Finally, we obtain the lateral thermal profile induced by a rectangular heater line $W \times L$ through superposition:

$$P(x) = \int_{-w/2}^{w/2} dx' \int_{-L/2}^{L/2} dy' G(r = \sqrt{(x-x')^2 + y'^2})$$

We carry out the 2D integration numerically using simple weighted summation of G over an adaptive, logarithmically spaced (x', y') rectangular mesh.

In diffusive regime, the solution can be derived fully analytically:

$$P_{\text{Fourier}}(x) = \frac{W}{8\pi D} \left[(\chi - 1) \ln \left(\frac{\sqrt{(\chi - 1)^2 + \beta^2} - \beta}{\sqrt{(\chi - 1)^2 + \beta^2} + \beta} \right) \right. \\
+ (\chi + 1) \ln \left(\frac{\sqrt{(\chi + 1)^2 + \beta^2} + \beta}{\sqrt{(\chi + 1)^2 + \beta^2} - \beta} \right) \\
\left. + 2\beta \ln \left(\frac{\sqrt{(\chi - 1)^2 + \beta^2} - (\chi - 1)}{\sqrt{(\chi + 1)^2 + \beta^2} - (\chi + 1)} \right) \right] \quad \chi = \frac{2x}{W}, \quad \beta = \frac{L}{W}$$

We set the line aspect ratio $\beta=40$ for the simulations in accordance with the fabricated samples, and then compute profiles for a variety of heater widths $0.1\mu\text{m} \leq W \leq 10\mu\text{m}$. We validated our computation scheme by observing excellent agreement between the numerical result for a tempered Lévy kernel with $\alpha = 1.999$ and the analytical Fourier solution (Figure R121).

Figure R121. Validation of integration scheme. Red circles: numerical result for tempered Lévy model with $\alpha = 1.999$; blue lines: analytical Fourier solution.

Figure R122 shows the tempered Lévy results. Similar to the experiment, the junction temperatures (within the heater line) increasingly exceed conventional predictions as the heater line gets narrower.

Figure R122. Tempered Lévy heating profiles and associated effective thermal performance for heater widths of 100nm, 1µm and 10µm.

However, contrary to measurement, the tails of the profile never go below the diffusive solution, but always stay slightly above. In other words, the thermal crosstalk at a nearby sensor (taken to be half a micron away from the line, i.e. $x = \frac{W}{2} + 0.5\mu\text{m}$) is either near nominal or even increased, but never reduced (Figure R123).

Figure R123. Tempered Lévy simulation of effective thermal performance at the heater ($x = 0$) and adjacent sensor half a micron away from the heater line ($x = \frac{W}{2} + 0.5\mu m$).

Failure of the tempered Lévy model to reproduce the measured tail anomalies is a direct and inevitable consequence of the fact that the associated Green's function nowhere crosses over with the diffusive counterpart (the response in the asymptotic Lévy regime, where $G \sim \frac{1}{r^{3-\alpha}}$, always exceeds the Fourier one). One should note that this failure would equally occur in BTE analysis within the relaxation time approximation (RTA). Indeed, the rigorous single pulse response of the 1D RTA-BTE invariably displays impeded thermal transport that gradually recovers (but never trumps) diffusive behaviour⁴; mathematical symmetry⁵ automatically conserves these traits in isotropic solutions of the multi-dimensional BTE.

Explaining the present measurement therefore clearly requires additional physics going beyond the current state-of-the-art RTA/superdiffusion formalisms. We should emphasize that the hydrodynamic model presented in the manuscript, can explain the reduced crosstalk near nanoheater sources due to the vortices in the heat flow. However, still a size-dependent thermal conductivity is needed to explain the increased heating at the source. A unified description including both hydrodynamic heat equation and tempered Lévy is not currently available and it is the subject of future research.

(3) Experimental evidence of hydrodynamic transport would be an important advance. But the current results are only mildly suggestive because no systematic study is conducted that tests the role of normal phonon-phonon scattering in nanoscale thermal transport.

We provided hydrodynamic model as a possible explanation for the observed temperature profiles. This provides better fits compared to previous models with less number of fitting variables. Also, we hope the new experimental thermoreflectance imaging results on amorphous SiO₂ films show that the observed reduced crosstalk is not an artefact of the thermoreflectance measurements at small scales.

We agree with the referee that the present results can be seen as an indirect way to show the presence of a nonlocal effect. There are many references describing nonlocal effects in thermal transport. The existence of a conservation law like the one associated in crystal momentum can be the main reason for it. We would like to highlight that our approach is not phenomenological but it can be deduced from the Boltzmann Transport Equation. In previous works it has been demonstrated how the nonlocal terms can be calculated from ab-initio results⁶⁻⁸. Additionally, we attached our upcoming paper showing that this nonlocal approach can also be used to explain the previous published results⁹. This works shows that a large number of experiments showing breakdown of Fourier law can be explained with a nonlocal hydrodynamic heat equation.

We also agree with the referee that the discussion about previous models and their shortcomings is important. Because of this, we have added the detailed response to questions 1 and 2 in the supplementary information (Sections S5).

(4) The hydrodynamic model appears to only predict vorticity within a few hundred nanometers of the heater line (Fig. 5). Because the Au temperature sensor cannot be perfectly adjacent to the heater line, this is exactly the spatial region the authors are not probing.

We respectfully disagree. Figure R141a and b, show a comparison between heat flux and temperature gradient in KCM and FEM models. It is clear that the misalignment between heat flux and temperature gradient near the top surface in KCM model extends beyond the neighbourhood of the heat source and reaches below the thermometer region. This in turn illustrate that the impact of the vorticity near heat source can be captured with thermal imaging and electrical measurements at the thermometer. This is evident in all thermal profiles for heaters that are smaller than ~0.5 microns in width in the sensor region ~1-2 microns away (see Figs. 3 and S4).

In addition, we should emphasize that the thermoreflectance data has sufficient spatial resolution to capture the temperature in the neighbourhood of the heater line on the oxide/substrate surface within few hundreds of nanometers. In these measurements, we independently calibrated the thermoreflectance coefficient on the oxide/substrate surface.

Figure R141. a. The Comparison between heat flux and $-\nabla T$ (negative gradient of temperature) vectors in KCM and Fourier models. b. The misalignment (angle difference) between heat flux and temperature gradient vectors adjacent to small sources. It is clear that the misalignment near the top surface in KCM model extends beyond the neighbourhood of the heat source and affects the thermometer region. This in turn illustrate that vorticity is not limited the region near heat source and can be captured with thermal imaging and electrical measurements by measuring the temperature of the thermometer.

(5) The authors don't validate their technique with a control experiment. They should show Fourier works for 255 nm heater lines on a material with a similar thermal conductivity in which they don't expect "quasiballistic transport"

This was a great suggestion that helped us better validate our measurements and modelling.

To address this concern, we fabricated a set of identical nanoheater lines on top of $3.5\mu\text{m}$ of SiO_2 and on silicon substrate. We measured several devices. Figure R151, summarize the results for $1\mu\text{m}$, 530nm and 300nm device sizes. The experimental results for these devices are compared with the FEM results. We used 1.41W/m-K for oxide and 124W/m-K for silicon thermal conductivity respectively, and $6.9\text{ nKm}^2/\text{W}$ for TBR between Au and SiO_2 in all the FEM simulations. The temperature profile both at the top of the heater line and on the neighbouring thermometer agree with the modelling which suggest no quasiballistic effect exist in these samples and a Fourier diffusive model can explain the entire temperature profile for different device sizes in an amorphous oxide material. These samples serve as control samples to validate thermoreflectance imaging technique.

Figure 151. Comparison between Experimental and modelling results on nanoheater lines on SiO_2/Si ($3.5\mu\text{m}/500\mu\text{m}$). a, e, i) optical images of $1\mu\text{m}$, 530nm , and 300nm heater lines. b, f, j) normalized temperature profile of the $1\mu\text{m}$, 500nm , and 300nm heater lines. c, g, k) Vertical cross section of temperature profile along y direction (along A-A' shown in Figure a) on top of the heater line. d, h, l) Horizontal cross section along the x direction (along B-B' shown in Figure e) on top of the thermometer. The red dot are experimental

results and the blue line is the modelling results. The temperature profiles both on top of the heater line and the neighbouring thermometer line agrees very well between experiment and modelling. The same conductivity values were used in all the modelling results (1.41W/m-K for oxide and 124W/m-K for silicon, and 6.9 nKm²/W for TBR between Au and SiO₂). These results show that the thermal transport in oxide is diffusive and a Fourier model can simply explain the entire temperature profile.

(6) The authors observe that the in-plane temperature profile is smaller than what the heat-equation predicts. This is consistent with the observations made in TDTR experiments from Cahill's group, see Fig. 1 of Ref. 24. The authors claim this is evidence of an "enhancement" of the heat-flow. I'm not entirely clear on what they mean by "enhanced". Experimentally, they only know the total heat-current out of the heater line, and the temperature profile perpendicular to metal line at the surface of the InGaAs. I don't see a way to know whether heat is travelling parallel or perpendicular to the surface. In other words, they aren't measuring an in-plane heat-current, so its unclear how they can conclude the in-plane heat current is greater or less than what Fourier theory would predict for the in-plane temperature profile they observe.

Thanks for your insightful comment. We agree with the referee that the usage of term "enhancement" or "enhanced heat flow" seems to be confusing and may not have been the correct terms to explain the observations made in the experiment.

By using the word "enhanced" we wanted to emphasize that as the size of the heater line decreases, the temperature rise in the neighbouring sensor is lower than the Fourier prediction (effective thermal conductivity higher than nominal).

To address this and avoid any confusion, we have changed the language in the text and substitute the terms "enhanced" to "lower thermal crosstalk" and "impeded" to "increased self-heating".

(7) Looking at Fig. 5, its difficult to see evidence of "enhancement" in the model calculation. In some positions, the heat-current is through-plane while the temperature gradient is in-plane. That seems like a suppression of the in-plane heat current. In other places, the heat-current is in-plane while the temperature-gradient is through-plane. I guess in those locations the in-plane heat-current is "enhanced". But on net, the heat-current must be suppressed, otherwise the temperature of the heater line would be lower than Fourier equation predicts, not higher.

Thanks again for your comment. As we described in question 6, By using the term "enhanced or enhancement" we would like to emphasize that as the size of the heater line decreases, the temperature rise in the neighbouring sensor is lower than the Fourier prediction (effective conductivity of the film is higher than nominal value). We believe the changes suggested in answering to question 6, that is to substitute the terms "enhanced" to "lower thermal crosstalk" and "impeded" to "increased self-heating", will address this question as well and resolve the corresponding ambiguities.

Also, the detailed comparison with anisotropic Fourier model (answer to question 1) should clarify the differences between purely diffusive and hydrodynamic (KCM) models. In the latter case, near the heater line, the Laplacian is non-zero and a "heat vorticity" appears.

(8) As the authors note in their supplemental information, recent work from both Cahill's group at Illinois, and Minnich's group at CalTech suggests that interfacial scattering impacts thermal transport in these types of problems. It doesn't appear the authors consider the role of the Au film or the interface in their experiments. How does the Hydrodynamic model account for interfacial scattering and the Au film? Since the transport is nonlocal, I don't think transport in the InGaAs can be treated independently from the transport across the Au/InGaAs interface.

Thanks for your comments. We believe part of our answer to question 1 regarding the thermal boundary resistance should answer this comment. We should emphasize that the heat is generated in nanoheater line using Joule effect and its amount could be characterized using electrical measurements independently. As the measurement is in steady-state and there are no ultrafast laser/material interactions, some of the complexities related to non-equilibrium between electrons and phonons could be avoided. Also, providing heat flux boundary condition on top of the InGaAs film (underneath the heater), one can calculate temperature profile near the heat source without worrying about unknown metal/semiconductor interface resistances. It is true that there could be still non-equilibrium phonon distributions (e.g. between acoustic or optical branches) very near the heat source. All we can say is that the thermoreflectance measurement of temperature on the semiconductor near the heater and on the metal sensor next to it, do not show any discontinuities. The thermoreflectance coefficient on different surfaces are calibrated independently by putting the sample on a temperature-controlled stage.

(9) The thermoreflectance value at 530 nm is determined by the electronic band structure of Au. The thermoreflectance near 530 nm in Au is high due to a d-band/s-band transition, see for example Eesley PRB 2144 (1986). The authors report a significant decrease in the thermoreflectance of the Au when they make the wire thin. It's difficult to see how 100 nm scale dimensions in the wire would affect the electronic structure of Au. It's more likely to be some sort of artifact related to the diffraction limit.

Thanks for your very interesting comment. It is true that the thermoreflectance coefficient of Gold as a material should not change significantly with sample size in 200-300nm width range. The effective thermoreflectance coefficient of heater line on the substrate is impacted by diffraction but its effect is taken into account in our models (the optical system has a known numerical aperture and this can be calibrated using small features in the SEM image). To verify that the blurring due to diffraction does not impact the comparison between theory and experiment, temperature measurements are performed at several different wavelengths in the visible range. An example is shown in Figure 191. Figure 191a and b shows the optical and temperature profile images of a 265nm heater

line within 500nm of a 2 μ m sensor line. The vertical cross section is plotted in Figure 191c. blue, green and red dots are the temperature measurements using 455nm, 530nm and 660nm LED lights. The black line is the FEM results using the effective thermal conductivity of 4.5W/m-K for InGaAs to match the temperature profile at the top. Figure 191d shows the expanded view of the temperature cross section on the sensor line. All the results producing the same observation that the temperature reduced on the thermometer by a factor of $\sim 2x$ compared to Fourier predictions suggesting that the observation is not an optical artefact.

Figure 191. a. Optical Image of a 265nm heater line along with a 2 μ m sensor line. b. Temperature profile at $I=6.2\text{mA}$. c. vertical Cross Section along a 265nm at 3 different wavelengths (455nm (blue), 530nm (green), 660nm (red)) compared with FEM (black line). d. Expanded view of the temperature profile cross section on the thermometer is plotted. It can be seen it is off by about 50%.

REVIEWER 2:

1. The captions of the figures are not informative. The authors should describe clearly what each panel in a figure means. For example, it is still not completely clear to the reviewer what was plotted in Fig. 4d. There are discussions of information of each panel in the text, but they are still not clear. The reviewer usually has to read the text many times to understand a figure.

Thank you very much. We updated the image captions for better clarity.

2. In Fig. 3a, for the small device, how to understand the spikes next to the heater from the hydrodynamics model?

The spikes that can be observed near the edges of the heater are due to the presence of a sharp corner (this is at the contact point of the heater and the substrate). This corner generates a large numerical rotational in the heat flux. These features are very typical in fluid simulations when corners/boundaries are present.

In the thermoreflectance measurements, we focus on the regions a little farther away from heaters. The temperature spikes are very localized and our temperature measurement are averaged over a region. The changes in the rest of the thermal map are not impacted by the sharp edges.

3. For the hydrodynamic model, is k isotropic? Will anisotropy influence the results? Why not make it anisotropic as it should be?

Thanks for your comment. Please see the detailed comparisons provided in the response to question 1 of reviewer 1. Here, we highlight some of the key points:

1. We need an anisotropic thermal conductivity of 1.5 W/mK in-plane and 10 W/mK cross-plane for InGaAs when the heater line is 265nm wide. These numbers are not physical for an epitaxially grown isotropic material such as InGaAs that is 5 μ m thick. Reduction of effective thermal conductivity has been observed for thin film materials, but this is typically for samples that are <0.5-1micron thin. Also, TDTR measurements on the same sample gave InGaAs thermal conductivity of ~6W/mK. It is hard to justify why the cross-plane thermal conductivity could be more than 50% larger and in-plane value 400% smaller.
2. For each nanoheater width on the same substrate a new pair of thermal conductivity values is required to fit the temperature profiles.

3. There are measurements that we could not find a pair of anisotropic thermal conductivity to fit the entire temperature profile
4. Even for the measurement in Figure R111 the thermal conductivity pair does not match the entire temperature profile and disagreement on the tails still exist
5. Hydrodynamic model does not need additional fitting parameter and anisotropic thermal conductivity assumption to explain the experimental results.

4. Fig. 4 shows temperature rise up to ~ 40 K, how will the temperature dependent thermal conductivity influence the temperature profile? Did the authors consider this? Non-local effect should also depend on temperature, since phonon MFP is a function of temperature.

Thanks for this important comments.

First, we would like to emphasize that in Figure 4 in the paper, while the average temperature change at the top of the metal is 40K, the temperature on the substrate is much lower and temperature dependence of InGaAs would not play a role. For example, in Figure 3b, while the temperature change at the top is about 35K, the tail is about 4K. The temperature on the InGaAs does not increase beyond 10K.

Second, the temperature-dependent thermal conductivity of InGaAs in 300-340K range is negligible (less than 5% change based on the TDTR data).

Third, Figure R241 below shows the temperature cross section of the a 10um heater line obtained from TR measurements at two different power levels of $P_1=57\text{mW}$ and $P_2=79.1\text{mW}$. The temperature cross section at power P_2 is shown in blue. If we normalize this temperature cross section with the ratio of P_1/P_2 we get the green curve which agrees within 1% with the temperature cross section at P_1 (Red curve). Temperature agreements at the top and the tail suggest that high temperature of the metal doesn't have significant effect on InGaAs and temperature dependent of InGaAs doesn't play a significant role.

Figure R241. a) temperature profile of a 10 μ m heater line. b) The vertical temperature cross section obtained from TR measurements at two different power level of $P_1=57\text{mW}$ (red curve) and $P_2=79.1\text{mW}$ (blue curve) is plotted. By normalizing the blue curve with the ratio of P_1/P_2 we get the green curve which is almost the same curve as is the red curve and shows the large temperature change at top of the heater line doesn't have any effect on the neighbouring devices where the temperature change is much smaller than the top.

5. The authors claims that Fourier's law with an effective thermal conductivity cannot describe the heat transfer correctly. However, thermal conductivity is a function of temperature as mentioned above and thus location. If such an effect is considered, will Fourier's law be potentially able to describe the heat conduction involving non-local effect? The reviewer feels that the authors should be more cautious in discounting Fourier's law as a way to describe non-local heat conduction, since there are so many studies out there showing that it can do such a job reasonably well.

As described in the response to the previous question, temperature-dependent thermal conductivity of InGaAs is negligible in the range of localized heating studied here. Also, measurements for different input powers don't show any non-linearity (see Fig. 2b in the manuscript and Fig. R241 in the previous question)

In addition, in response to question 1 of reviewer 1, we compared several methods in the literature that are available to explain the data using the Fourier model and highlighted some of their shortcomings.

6. The resolution of the TRI method is 200-300 nm. The smallest device has a feature length of 263 nm, which is similar to the resolution. How would this influence the accuracy of temperature measurement and thus reliability of the data.

This is an important comment. All our measurements are independently verified using electrical measurements and we don't rely on only the TR data. This is shown in Figure 2b of the paper. For the comparisons in thermoreflectance images, we focus on sensor lines that are bigger than the diffraction limit (see Fig. 3b).

For your reference, we have plotted the temperature measurement of 10 μ m and 265nm heater lines at different current levels using both electrical and thermoreflectance measurement in Figure R261.

In addition, in an upcoming paper we are showing that image processing techniques can be used to simulate the effect of diffraction, and we are able to extract temperature profile of devices well below diffraction limit (down to 100-200nm) even without independent electrical measurements. The conference paper for this work is already published in IEEE SemiTherm 2015¹⁰ and IEEE ITherm 2017¹¹.

Figure 1. a) temperature profile of a 10 μ m heater line. b) Average temperature on top of the heater line at different current level using Thermal Imaging (red), four probe electrical measurement (green) and Fourier finite element model (black line). c) temperature profile of a 265nm heater line. d) Average temperature on top of the heater line at different current level using Thermal Imaging (red), four probe electrical measurement (green) and Fourier finite element model (black line).

References

1. Minnich, A. J. *et al.* Thermal conductivity spectroscopy technique to measure phonon mean free paths. *Phys. Rev. Lett.* **107**, 1–4 (2011).
2. Siemens, M. E. *et al.* Quasi-Ballistic thermal transport from nanoscale interfaces observed using ultrafast coherent soft X-ray beams. *Nat. Mater. Lett.* **9**, 26–30 (2010).
3. Vermeersch, B. Compact stochastic models for multidimensional quasiballistic thermal transport. *J. Appl. Phys.* **120**, (2016).
4. Hua, C. & Minnich, A. J. Transport regimes in quasiballistic heat conduction. *Phys. Rev. B - Condens. Matter Mater. Phys.* **89**, (2014).
5. Hua, C. & Minnich, A. J. Analytical Green's function of the multidimensional frequency-dependent phonon Boltzmann equation. *Phys. Rev. B - Condens. Matter Mater. Phys.* **90**, 1–7 (2014).
6. de Tomas, C., Cantarero, a., Lopeandia, a. F. & Alvarez, F. X. From kinetic to collective behavior in thermal transport on semiconductors and semiconductor nanostructures. *J. Appl. Phys.* **115**, 164314 (2014).
7. Torres, P. *et al.* First principles kinetic-collective thermal conductivity of semiconductors. *Phys. Rev. B* **95**, 1–5 (2017).
8. de Tomas, C., Cantarero, a, Lopeandia, a F. & Alvarez, F. X. Thermal conductivity of group-IV semiconductors from a kinetic-collective model. *Proc. R. Soc. A Math. Phys. Eng. Sci.* **470**, 20140371–20140371 (2014).
9. Torres, P. *et al.* Experimental and theoretical study of collective hydrodynamic heat transport at nanoscales. *Submitted* (2017).
10. Ziabari, A. *et al.* Sub-diffraction Limit Thermal Imaging for HEMT Devices. *31th Annu. IEEE Therm. Meas. Model. Manag. Symp.* 1–6 (2015).
11. Ziabari, A. *et al.* Sub-Diffraction Thermoreflectance Thermal Imaging using Image Reconstruction. in *16th IEEE Conference on Thermal and Thermomechanical Phenomena in Electronic Systems (IITHERM)* 122–128 (2017).

Reviewers' Comments:

Reviewer #1:

Remarks to the Author:

Overall, the manuscript is much improved. The authors provided detailed responses to all my original concerns. I recommend the manuscript for publication in Nature Communications.

Reviewer #2:

Remarks to the Author:

I thank the reviewer for addressing my (Reviewer 2) comments. I am generally satisfied with the responses. However, I would like the authors to implement these into the manuscript as much as possible since the readers may raise the same questions. Other than that, I think the paper is good to go.